Minimizing features while maintaining performance in data classification problems

Matharaarachchi Surani 1 matharas@myumanitoba.ca
http://orcid.org/0000-0001-9129-6676 Domaratzki Mike 2
Muthukumarana Saman 1
1 Statistics, University of Manitoba , Winnipeg, Manitoba , Canada
2 Computer Science, University of Western Ontario , London, Ontario , Canada
Shang Yilun
Electronic publication date: 2022 Sep 14
Publication date: 2022
Volume: 8
Electronic Location ID: e1081
Received 2022 Mar 2; Accepted 2022 Aug 10
Copyright: © 2022 Matharaarachchi et al.
Copyright year: 2022
Copyright holder: Matharaarachchi et al.
License: This is an open access article distributed under the terms of the Creative Commons Attribution License, which permits unrestricted use, distribution, reproduction and adaptation in any medium and for any purpose provided that it is properly attributed. For attribution, the original author(s), title, publication source (PeerJ Computer Science) and either DOI or URL of the article must be cited.
License URL: https://creativecommons.org/licenses/by/4.0/

Keywords: Feature selection, Principal component loading, Classification, Class imbalance

Funding: Natural Sciences and Engineering Research Council of Canada Saman Muthukumarana has been supported by research grants from the Natural Sciences and Engineering Research Council of Canada. There was no additional external funding received for this study. The funders had no role in study design, data collection and analysis, decision to publish, or preparation of the manuscript.

==============================
High dimensional classification problems have gained increasing attention in machine learning, and feature selection has become essential in executing machine learning algorithms. In general, most feature selection methods compare the scores of several feature subsets and select the one that gives the maximum score. There may be other selections of a lower number of features with a lower score, yet the difference is negligible. This article proposes and applies an extended version of such feature selection methods, which selects a smaller feature subset with similar performance to the original subset under a pre-defined threshold. It further validates the suggested extended version of the Principal Component Loading Feature Selection (PCLFS-ext) results by simulating data for several practical scenarios with different numbers of features and different imbalance rates on several classification methods. Our simulated results show that the proposed method outperforms the original PCLFS and existing Recursive Feature Elimination (RFE) by giving reasonable feature reduction on various data sets, which is important in some applications.

Introduction

With the increased development of machine learning concepts and related topics, feature selection has become crucial as most real-world data sets suffer from many features. This problem is known as the curse of dimensionality (Bellman, 1957), and many sectors negatively experience this issue, including in the worlds of business, industry, and scientific research.

Selecting fewer features, known as feature selection, provides several significant advantages. With feature selection, dimensionality reduction can decrease the size of the data without harming the overall performance of the analytical algorithm (Nisbet, 2012). The decrease of computational time while increasing the algorithm’s predictive power and interpretability are notable gains (Miche et al., 2007; Samb et al., 2012). Then again, a model with fewer features may be more interpretable and less costly, especially if there is a significant cost of measuring the features. Statistically, it is more convenient and attractive to estimate fewer parameters, and it will also reduce the negative impact of non-informative features. Further, it becomes increasingly challenging to reveal patterns in data with many features (Guo et al., 2002).

The main three categories of feature selection techniques are filter, wrapper, and embedded methods. Filter methods measure the feature relevance to the dependent variable; hence, only features with meaningful relationships would be included in a classification model. They use statistical methods such as Pearson’s Correlation, Analysis of Variance (ANOVA), Linear discriminant analysis (LDA), and Chi-Squared statistics to select a subset of features. By training a model, wrapper methods measure the usefulness of a subset of features (Saeys, Inza & Larrañaga, 2007). Forward Feature Selection, Backward Feature Elimination (Weisberg, 2005), and Recursive Feature Elimination (RFE) (Guyon et al., 2002) are typical examples of commonly used wrapper methods. The third category, embedded methods, optimize the objective function or performance of a learning algorithm or model and also use an intrinsic model-building metric during learning. L1 (LASSO) regularization (Tibshirani, 1996) and Elastic Net (Zou & Hastie, 2005) are commonly known embedded methods. Combining these three types of techniques to produce ensemble feature selection is called the ensemble feature selection method, which combines multiple feature subsets to select an optimal subset of features. Hashemi, Dowlatshahi & Nezamabadi-pour (2021) has proposed a multi-criteria decision-making (MCDM) approach, which is an ensemble of filter methods. This article mainly considers the wrapper methods (Kohavi & John, 1997), which iteratively examine different subsets to improve accuracy on fewer features. RFE (Guyon et al., 2002) is one such commonly used technique. In standard RFE, a feature is eliminated if it is the least important to predicting, and features are ranked according to the model’s strength by considering the performance scoring method.

Various approaches and extensions in the literature have been suggested to the existing feature selection mechanisms such as RFE. Samb et al. (2012) introduced an RFE-SVM-based feature selection approach by reusing previously removed features in RFE. They have used two local search tools, Bit-Flip (BF) and Attribute-Flip (AF), to improve the quality of the RFE. But this approach is specific to the SVM classification, where our suggested method can be applied to any classification method, which facilitates a feature ranking criterion with feature importance. An enhanced recursive feature elimination has been introduced by Chen & Jeong (2007) which is also an algorithm based on RFE and SVM. It also assesses a weak feature removed by the standard RFE based on the classification performance before and after removing that feature and reconsidering it in the feature subset. There are other proposed methods that use thresholds to identify the feature subset. A ROC-based feature selection metric for small samples and imbalanced data (FAST) is recommended by Chen & Wasikowski (2008). This method is based on the area under a ROC curve by discretizing the distribution. An extension of the FAST method, but another threshold-based feature selection (TBFS) technique is discussed by Wang, Khoshgoftaar & Van Hulse (2010), where they produce 11 distinct versions of TBFS based on 11 different classifier performance metrics. A cluster-based feature selection, SVM-RCE, has been introduced by Yousef et al. (2007, 2021), which uses K-means to identify correlated gene clusters and SVM to identify the ranks of each cluster. Then, the recursive cluster elimination (RCE) method iteratively removes the clusters with the least performance accuracy.

Usually, the two main objectives for feature selection are to select the smallest possible subset with a given discrimination capability and to find the subset of features with the minimum possible generalization error (Granitto et al., 2006). This article examines different subsets of features to maintain accuracy on fewer features.

We propose a method as an extended version of the suggested PCLFS (Principal Component Loading Feature Selection) method (Matharaarachchi, Domaratzki & Muthukumarana, 2021; Matharaarachchi, 2021) explained in “Methods and Experimental Design”. PCLFS, a wrapper-based feature selection technique, ranks features by the sum of absolute values of principal component loadings. After determining the order of the importance of each feature and obtaining accuracy measures for each subset, the remaining question in feature selection is how to determine the best number of features. PCLFS uses a conventional feature selection method, the sequential forward selection, to choose the optimal feature subset. It fits a model and captures the most informative feature subset, which is the subset that maximizes the F1-score using a sequential feature selection technique. By adding one or a small number of features per loop, PCLFS attempts to eliminate dependencies and collinearity in the model. The proposed method further identifies a local maximum with a practical implication. Several other optimization mechanisms also have been introduced in the literature to search for the optimal feature subset. Out of many such methods, Particle swarm optimization (PSO)-based feature selection (Kennedy & Eberhart, 1995; Shi & Eberhart, 1998), which is a kind of heuristic algorithm based on swarm intelligence, has gained significant attention. This algorithm finds the optimal solution through collaboration and information sharing between individual groups of features. In “Experimental Results”, we will also compare our results with some PSO-based methods.

Prior research also compares the impact of class re-balancing techniques on the performance of binary prediction models for a different choice of data sets, classification techniques, and performance measures. Hence, in this article, we focus on binary classification problems with only two possible outcomes. Class imbalance occurs when the number of instances in the small (minority) class is significantly smaller than that in the large (majority) class. It produces a significant negative influence on standard classification learning algorithms. The minority class is important in many practical situations; therefore, it requires an intense urgency to be identified (Sun, Wong & Kamel, 2009). However, studies on class imbalance classification have gained more emphasis only in recent years (Kotsiantis, Kanellopoulos & Pintelas, 2005) and many re-sampling methods have been introduced to eradicate this issue. This article will mainly use Synthetic Minority Oversampling TEchnique (SMOTE) (Chawla et al., 2002) as a re-balancing technique to achieve higher accuracy in applications.

Problem statement

Although we can already reduce the number of features using PCLFS according to a given selection scoring criteria, there is room to improve it further. We observed that the number of features of the chosen subset by PCLFS might not be the expected quantity if the desire is to have a smaller number of features. In particular, there are other selections of a lower number of features with negligibly lower model accuracy. Therefore, we consider the challenge of finding an optimal threshold to identify this minuscule difference. We also compare simulation and application results with existing PCLFS and RFE results.

To illustrate this procedure with a contrived example we will consider a simulated data set with ten informative features out of 30. Figure 1 shows an ideal PCLFS curve; the curve leaps to an excellent accuracy when the 10 informative features are captured, then slightly decreases F-score as the non-informative features are added into the model.

Figure 1 Ideal Feature selection using PCLFS method.

The dotted line indicates the actual number of informative features. The red point indicates the PCLFS feature selection, which selects all the informative features in the data set.

But, example 1 in Fig. 2 shows a plot of the F1-score of different sized subsets of a fixed data set, all chosen based on the PCFLS method described in “Methods and Experimental Design”. This figure shows that PCLFS (blue point) has selected 29 features, but the F1-score does not appear to be much improved after around 10 features. Meanwhile, the proposed method (red point) suggests 10 features as the smaller number of features with similar performance. According to Fig. 3 (example 2), the proposed approach (red point) finds a comparable value to the informative features in the data set under the given threshold, while the original selection (in blue) is far away from the desired number of features to be selected.

Figure 2 Example 1: Feature selection using proposed method.

The dotted line indicates the actual number of informative features. The red point indicates the PCLFS feature selection with number of selected features and the F1-score while the red point explains the same for the proposed method.

Figure 3 Example 2: Feature selection using proposed method.

The dotted line indicates the actual number of informative features. The red point indicates the PCLFS feature selection with number of selected features and the F1-score while the red point explains the same for the proposed method.

Goal

Our primary focus in this article is analyzing the behavior of the PCLFS method towards classification accuracy and suggesting an improved extension for selecting a smaller number of features with similar performance with the previous method. Hence, we introduce an algorithm with a threshold to achieve this objective. Besides choosing the minimal number of features, we suggest the appropriate feature subset by considering the informativeness of features. To cover most practical scenarios, we synthetically simulated data using the scikit-learn python library (Pedregosa et al., 2011) and compared the performance of the existing and the proposed method. These algorithms will be further examined on five benchmark continuous data sets with different numbers of objects, imbalance rates, and features to derive further conclusions. For the practical scenario, we also use a re-sampling technique, Synthetic Minority Over-sampling Technique (SMOTE) (Chawla et al., 2002) to determine the performance of the model on the imbalanced data set.

The remainder of this article is structured as follows. Section 2 describes the data preparation introduces the methods used in the study with the experimental design. Section 3 presents the results of the simulation studies, and the results in a real-world application are illustrated and interpreted in Section 4. Finally, Section 5 of this article is included with a discussion of its contributions and limitations.

Methods and experimental design

RFE

RFE can be fitted on any classification model with an inherent quantification of the importance of a feature. It removes the weakest features by a step count, where the step is the number of features removed at each iteration. This process repeats until the stipulated number of features is reached. Features are ranked according to the importance identified by the model. Then, to find the optimal number of features, cross-validation is used in each iteration and selects the subset giving the best scoring value as the desired feature subset.

PCLFS

Principal Component Loading Feature Selection uses the sum of absolute values of principal component loadings to order features and capture the most informative feature subset, which is the subset that obtains the maximum F1-score using a sequential feature selection technique (Matharaarachchi, Domaratzki & Muthukumarana, 2021). This method can be fitted on any classification model as feature ordering is entirely independent of the classification method. The PC (principal components) loadings are the coefficients of the linear combination of the original variables constructed by the PCs. In this study, PCLFS orders features using the sum of the first two PC loadings’ absolute values, trains classification models on training data, and selects the optimal feature subset that obtains the maximum F1-score. Starting from the most informative feature, it adds features one by one according to the order defined by the sum of the first two PC loadings until all features are added. Hence the total number of subsets will equal the number of features in the data set. It does testing at each step (i.e., F1-score) and, in the end, obtains the feature subset which gives the maximum F1-score.

inputs:

Training samples: X0=[X1,X2,…,Xℓ]T

Class labels: y=[y1,y2,…,yℓ]T

outputs:

Feature ranked list: r=[r1,r2,…,rn]

Grid scores: g=[g1,g2,…,gn]

Number of selected features by PCLFS: npclfs

Here, n is the number of features in the data set, and ℓ is the number of samples in the training set. Grid scores (g) are the F1-scores such that gi corresponds to the F1-score of the ith feature subset with the first i features of the PCLFS ordered feature list.

PCLFS is a newly introduced feature selection method (Matharaarachchi, Domaratzki & Muthukumarana, 2021). Therefore, we use RFE to compare results as RFE is one of the most commonly used wrapper feature selection methods.

Suggested method

In this article, we propose a new algorithm based on PCLFS. The suggested method is an extension of the PCLFS method, and the results that come out of the PCLFS algorithm are fed into the new algorithm to get the desired output. The main difference between the new method and the original PCLFS is that the original PCLFS chooses the feature subset giving the maximum score. In contrast, the suggested method identifies a feature subset under an applicable threshold to obtain a smaller feature subset with similar performance and minimal loss. We compare PCLFS and the extended method on various synthetic data sets and show that the suggested method reduces the number of features with a bearable score reduction. The algorithm for the new method is described below.

inputs:

Grid scores: g=[g1,g2,…,gn]

Number of selected features by PCLFS: npclfs

Total number of features: n

Feature importance scores (obtained from the classifier): i=[i1,i2,…,inpclfs]

Maximum tolerable F1-score reduction: T (User-defined).

procedure: Step 1: Consider all the local maximum grid scores ( gj) corresponding to the number of subsets of features selected by PCLFS which is less than the optimal number of features selected ( npclfs) where,

gj>max(gj−1,gj+1),j<npclfs

Step 2: Connect each point with the maximum point ( gnpclfs) and compute each line’s gradient values (i.e., the tangent value of the cone).

Step 3: Compare the gradient values with a threshold value t.

(1) gradient=(Δy)j(Δx)j<t

The threshold (t) can be interpreted as the tolerable reduction of the F1-score to reduce one feature, where,

(2) t=MaximumtolerableF1scorereductionTotalnumberoffeatures=Tn

Step 4: Obtain the F1-score, which gives the smallest number of features ( nproposed).

Note: If there is no value found for the given condition, we will return the same PCLFS results.

Step 5: To get the relevant feature subset, use feature importance scores (i). Then obtain the best nproposed features as the smallest feature subset with similar performance (s).

outputs:

The smallest number of features with minimum scoring loss: nproposed

Relevant feature subset: s.

Figure 4 presents how the algorithm picks the desired selection using the gradient method. In our algorithm, if we only consider F1-scores that give the smaller number of features, sometimes we end up with values where the neighbors are larger, and a larger F1-score for the neighbor indicates that the neighbor should be chosen. To avoid such situations and be well-defined, we require the selected value to be a local maximum besides having the smallest number of features.

Figure 4 Visualization of the hypothetical execution of the proposed algorithm.

Graphical view of the suggested algorithm. θj is the angle between the horizontal dotted line (a line parallel to the number of selected features axis) and the red line, which combines the jth point with the maximum point. The blue point indicates the PCLFS feature selection with number of selected features and the F1-score while the red point explains the same for the proposed method.

Finding an optimal threshold to distinguish the small difference between F1-scores was challenging as it depends on many factors. Therefore, the gradient method was introduced to find the F1-score reduction per feature for each subset selection. We compared each gradient with the maximum bearable gradient value. When we only consider a numerical cut-off value as the threshold, it will reduce the same amount regardless of the number of features removed. The tolerable F1-score should be explained for a single feature reduction to avoid this problem. We also observed that when the number of features in the data set increases, F1-score reduces drastically unless the threshold is extremely small, and it is required to change the threshold according to the number of features in the data set. Therefore, the threshold had to be defined to include the number of features as a parameter to have consistent solutions. Hence, we considered a tolerable F1-score decrease for one feature, in other words, “the threshold,” by having the maximum tolerable F1-score reduction over all the features.

Simulation study

When introducing an algorithm, we performed a simulation study to determine how the factors affect the behavior of the final result. Therefore, we synthetically simulate samples, where the sample size is 1,000. The number of classes is two (binary classification), and there is only one cluster per class. Several numbers of features were considered to compare different situations. Since different classification models perform uniquely in different data sets, we aim to introduce a general tool that works with multiple models. We train different binary classification models in data sets with different numbers of features and imbalance rates to ensure this. Initially, five different binary classification models were trained with PCLFS. They are Logistic Regression (LOGIT) (Weisberg, 2005), Linear Support Vector Machine (SVM_Linear) (Xia & Jin, 2008), Decision Tree, Random Forest (RFC) (Breiman, 2001), and Light Gradient Boosting Classifier (LGBM_C) (Friedman, 2001).

Simulation results

This section illustrates the results obtained through synthetic samples and the simulation study results on all three methods, existing RFE, PCLFS, and proposed PCLFS-ext.

To capture the variability of the final F1-scores of each method, we conducted a simulation study to determine the validity of the suggested combined approach. One hundred samples are simulated from each scenario to reduce the variability in experimental results, while the number of informative features is increased from one to the total number of features. All features are classified as informative or non-informative. No redundant features or repeated features are included in simulated data sets. We generated data for 50%:50% balanced data and two other imbalance rates, 70%:30% and 90%:10%. Two sample sizes with 200 and 1,000 samples were also examined, and results were discussed only for sample size 200 unless there is a notable discrepancy to emphasize. Most importantly, in this analysis, the models were fitted on original data and re-sampled data with SMOTE. Here, the results are only illustrated for the logistic regression model. Supplemental Materials contain results for other classification models, with highly imbalanced data with a 90%:10% rate and a sample size of 1,000.

Simulation results without SMOTE

Results obtained for the comparison of model F1-scores and feature selection correct percentages of RFE and PCLFS are shown in Fig. 5. The figure shows results for sample sizes of 200 for the Logit classifier when the threshold is 0.0017. However, we also compare the extended version of PCLFS (PCLFS-ext), which gives even a higher feature selection correct percentage for an insignificantly smaller F1-score reduction over the PCLFS method.

Figure 5 Simulation results for Logit-200 sample size (Without SMOTE).

Final model F1-scores and feature selection correct percentages for the Logit model, when the threshold is 0.0017.

To further understand the selection of features, we plotted the number of selected features and feature selection true positive rate ( TPRfs) against the number of informative features given. Feature selection TPR was calculated using the equation explained in Matharaarachchi, Domaratzki & Muthukumarana (2021). For the original data, PCLFS and PCLFS-ext methods pick a relatively larger number of features than RFE. Nevertheless, the feature selection TPR is significantly higher in the proposed methods. The results with 200 sample size are shown in Fig. 6. We note that when the sample size is smaller, the PCLFS-ext method is not tempted to pick a lower number of features in highly imbalanced data under the given threshold of 0.0017. But, for a higher sample size of 1,000, the proposed method outperformed the existing methods in each scenario considered in the simulation. In Fig. 7 the results are shown for an imbalance rate of 0.9:0.1. Similar but high pronounced effects are visible at the other imbalance rates.

Figure 6 Simulation results for Logit-200 sample size (Without SMOTE).

Number of selected features and feature selection TPR for the Logit model, when the threshold is 0.0017.

Figure 7 Simulation results for Logit-1,000 sample size (Without SMOTE).

Accuracy measures for the Logit model for an imbalance rate of 0.9:0.1, when the threshold is 0.0017.

Simulation results with SMOTE

We repeated the same procedure for imbalanced data by re-balancing using SMOTE with the Logit classifier for sample sizes 200 and 1,000. Except for having lower feature selection correct percentages for highly imbalanced data with smaller sample sizes (in Fig. 8), in all the other scenarios, PCLFS extended version performs much better than PCLFS and RFE on the same data set. Meanwhile, for data sets with a larger sample size (e.g., 1,000), PCLFS and PCLFS-ext methods even pick a lower number of features than RFE when there are few informative features in the data set (Fig. 9). This property is valuable when we are dealing with real-world problems.

Figure 8 Simulation results for Logit-200 sample size (With SMOTE).

Accuracy measures for the Logit model for an imbalance rate of 0.9:0.1, when the threshold is 0.0017.

Figure 9 Simulation results for Logit-1,000 sample size (With SMOTE).

Accuracy measures for the Logit model for an imbalance rate of 0.9:0.1, when the threshold is 0.0017.

Experimental results

SPECTF heart data

To analyze the behavior of models on a real-world data set, we consider the publicly available Single-photon emission computed tomography (SPECT) heart data set (Kurgan et al., 2001; Krzysztof, Daniel & Ning, 1997; Bache & Lichman, 2013), which describes diagnosing cardiac abnormalities using SPECT. This is the same data set used in (Matharaarachchi, Domaratzki & Muthukumarana, 2021), and use it in order to be consistent with the analysis and results. Response of the data set consists of two categories: normal and abnormal, by considering the diagnosis of images. This data consists of binary class imbalanced data with a higher number of numerical features and a lower number of instances.

The sample consists of data from 267 patients with 44 continuous features that have been created for each patient. Hence, it has 267 instances that are described by 45 attributes (44 continuous and one binary class). We also divided the data set into two groups, 75% training samples and 25% test samples. The class-imbalanced rate for the data set is 79.4%:20.6%, where the minority class represents the abnormal patients. The imbalance is the same in the training and test set.

Then we applied Synthetic Minority Oversampling Technique (SMOTE) to handle imbalanced data to achieve higher accuracy in classification models. The SMOTE aims to balance class distribution by randomly increasing minority class examples by creating similar instances.

We compare the proposed PCLFS and PCLFS-ext model results with the final F1-scores of the existing RFE method. The results are shown in Table 1 highlighting the best results. For SMOTE data, PCLFS selects a smaller number of features than RFE, with a higher F1-score for all the classification models. It further reduces the number of features considerably in the PCLFS-ext method for the Logit, decision tree, and RFC models, and the last two columns of the Table 1 depicts the reduction/increment of the percentages of features and the F1-scores over RFE and the proposed method where

Table 1 Final F1-score comparison between RFE and proposed methods (PCLFS/PCLFS-ext (t = 0.0011)).

Bold font indicates the best result based on the F1-score.

SMOTE	Method	Basic	RFE	PCLFS	PCLFS-ext	Feature reduction%/(increment%)	F1-score (reduction)/increment	
		#Features	F1-scores	#Features	F1-scores	#Features	F1-scores	#Features	F1-scores	
TRUE	Logit	44	0.6809	36	0.6957	24	0.6957	11	0.6939	56.8%	(0.0018)	
	LGBM	44	0.6667	27	0.6286	13	0.7027	–	–	31.8%	0.0741	
	Decision Tree	44	0.5556	44	0.5556	9	0.6667	3	0.6666	93.2%	0.1110	
	RFC	44	0.6486	38	0.6111	42	0.7059	12	0.6842	59.0%	0.0731	
	SVM-Linear	44	0.6511	30	0.6977	12	0.7727	–	–	40.9%	0.0750	
FALSE	Logit	44	0.5455	30	0.5000	44	0.5455	–	–	(31.8%)	0.0455	
	LGBM	44	0.6250	15	0.5455	15	0.6250	–	–	0.0%	0.0795	
	Decision Tree	44	0.5294	27	0.5161	9	0.5946	–	–	40.9%	0.0785	
	RFC	44	0.2609	9	0.3704	11	0.4444	–	–	(4.5%)	0.0740	
	SVM-Linear	44	0.5946	21	0.5882	37	0.6316	–	–	(36.4%)	0.0434	

Featurereduction/(increment)%=Numberoffeaturesreduced/(increased)Totalnumberoffeatures.

Figure 10 displays how the PCLFS-ext version picks a lesser number of features with similar performance with a maximum tolerable F1-score of 0.05, hence the threshold of 0.0011. Similar to the simulation results in “Simulation Results”, the PCLFS-ext method picked a lower number of features than PCLFS when the data set is balanced.

Figure 10 Number of features selected by each method.

Selecting smaller number of features under the threshold of 0.0011. The red point indicates the PCLFS selection whereas the blue point indicated the PCLFS-ext method selection.

Further experiments on different data sets

To further evaluate the performance of the existing and proposed approaches, we used five different continuous data sets which downloaded from UCI machine learning repository (Bache & Lichman, 2013). They all have a binary response variable with a different number of cases, features and imbalance rates (Table 2). For every trial, we divided each data set into two groups, 75% training samples and 25% test samples. To capture the variability of imbalance data, we executed methods with and without SMOTE on the same data sets. The Logit model was used as the classifier and classification error rate and F1-score were used to evaluate the performance of each method on all data sets.

Table 2 Continuous data sets.

Data set	Number of features	Number of instances	Number of classes	Class imbalance rate	
German	24	1,000	2	(0.7:0.3)	
Ionosphere	34	351	2	(0.73:0.27)	
SPECTF	44	267	2	(0.79:0.21)	
Sonar	60	208	2	(0.53:0.47)	
Musk-Version 1	166	476	2	(0.57:0.43)	

Table 3 indicates the comparison of different methods after 50 independent trials on each data set. Here, ‘Basic’ is the data set with the original feature set utilized for classification. ‘Size’ indicates the average number of features selected by each method in 50 independent trials. Other than having F1-score, we used classification accuracy (error rate) to compare performance. ‘Best,’ ‘mean,’ and ‘std dev’ implies the best, the average, and the standard deviation of the classification error.

Table 3 Result of existing RFE, proposed PCLFS-ext, and PSO-based methods proposed by Huda & Banka (2022) on continuous data sets.

Bold font indicates the best result for each data set based on the error rate.

Data sets	Methods	Without SMOTE	With SMOTE	
		Size	Best	Mean ± Std_dev	F1-score	Size	Best	Mean ± Std_dev	F1-score	
SPECTF	Basic	44.00	11.89	24.99 ± 6.01	0.3961	44.00	12.59	27.57 ± 6.44	0.4353	
	PCLFS	37.14	11.89	19.08 ± 3.93	0.4475	15.18	11.89	23.05 ± 5.24	0.5818	
	PCLFS-ext	34.22	12.59	22.59 ± 4.68	0.4457	11.70	11.89	26.15 ± 5.9	0.5794	
	RFE	21.94	12.59	24.57 ± 4.92	0.3783	28.54	11.19	27.49 ± 6	0.4526	
German	Basic	24.00	19.33	23.99 ± 1.89	0.8380	24.00	22.67	26.62 ± 1.84	0.8118	
	PCLFS	17.62	18.67	23.17 ± 1.86	0.8445	18.52	21.33	25.27 ± 1.75	0.8214	
	PCLFS-ext	4.08	23.33	27.69 ± 2.35	0.8281	13.80	21.33	25.76 ± 1.99	0.8155	
	RFE	17.88	19.33	24.5 ± 2.49	0.8320	21.26	21.00	26.83 ± 2.11	0.8104	
	PSOFRFSA	16.14	21.78	22.18 ± 1.31						
	PSOFRFSAN 0.9	7.9	19.02	21.17 ± 1.67						
	PSOFRFSAN 0.5	5.47	19.38	21.91 ± 1.07						
	PSOFRFSANA 0.9	7.81	19.02	21.01 ± 1.57						
	PSOFRFSANA 0.5	5.41	19.38	21.37 ± 1.37						
Ionosphere	Basic	34.00	6.60	12.74 ± 2.84	0.9033	34.00	8.49	14.04 ± 2.68	0.8940	
	PCLFS	30.32	6.60	12.28 ± 2.68	0.9073	30.24	8.49	13.32 ± 2.21	0.8978	
	PCLFS-ext	26.82	6.60	12.79 ± 3.04	0.9038	25.62	8.49	13.79 ± 2.61	0.8940	
	RFE	18.18	6.60	13.25 ± 3.14	0.9017	22.92	9.43	14.38 ± 2.69	0.8918	
	PSOFRFSA	19	5.18	6.81 ± 3.13						
	PSOFRFSAN 0.9	4	5.49	6.84 ± 4.1						
	PSOFRFSAN 0.5	3.7	5.39	6.93 ± 3.93						
	PSOFRFSANA 0.9	3.7	5.39	6.94 ± 3.27						
	PSOFRFSANA 0.5	3.7	5.39	6.98 ± 3.19						
Sonar	Basic	60.00	15.87	24.44 ± 4.94	0.7228	60.00	14.29	24.22 ± 4.86	0.7416	
	PCLFS	39.60	14.29	22.7 ± 4.58	0.7309	39.54	14.29	22.06 ± 4.1	0.7599	
	PCLFS-ext	38.86	14.29	22.76 ± 4.65	0.7304	38.00	14.29	22.1 ± 4.07	0.7589	
	RFE	17.04	15.87	25.27 ± 6.3	0.7104	15.34	14.29	24.54 ± 5.42	0.7467	
	PSOFRFSA	34	17.04	19.4 ± 4.01						
	PSOFRFSAN 0.9	7	14.77	15.2 ± 6.27						
	PSOFRFSAN 0.5	76.71	15.08	16.78 ± 5.45						
	PSOFRFSANA 0.9	6.02	14.01	15.93 ± 4.2						
	PSOFRFSANA 0.5	5.13	14.97	15.79 ± 4.02						
Musk-Version1	Basic	166.00	10.49	17.68 ± 3.22	0.8128	166.00	10.49	17.61 ± 2.53	0.8121	
	PCLFS	149.22	10.49	15.61 ± 2.68	0.8315	150.86	9.79	15.76 ± 2.19	0.8301	
	PCLFS-ext	141.90	10.49	15.8 ± 2.8	0.8302	143.73	9.79	15.97 ± 2.33	0.8286	
	RFE	73.88	13.29	18.94 ± 4.74	0.7879	88.65	10.49	19.06 ± 3.49	0.7945	
	PSOFRFSA	95.71	22.15	23.11 ± 3.01						
	PSOFRFSAN 0.9	37.77	22.78	24.12 ± 3.42						
	PSOFRFSAN 0.5	37.77	22.78	23.19 ± 3.47						
	PSOFRFSANA 0.9	37.7	21.19	22.51 ± 4.01						
	PSOFRFSANA 0.5	36.17	20.17	21.91 ± 3.97						

Table 3 depicts that our proposed method outperforms the existing RFE feature selection method in various data sets by accomplishing equivalent or higher accuracy. We also cross-checked the results of the proposed method with the results obtained by Huda & Banka (2022) for different PSO-based feature selection methods while using the same real-world data sets, German, Ionosphere, Sonar, and Musk-Version1. These methods include some existing PSO feature selection methods such as PSOPRS, PSOPRSN ( α=0.9 and α=0.5), and PSOPRSE, and some newly proposed efficient feature selection methods using PSO with the fuzzy rough set as fitness function (PSOFRFSA, PSOFRFSAN, and PSOFRFSANA). Result of proposed methods by Huda & Banka (2022) on the same continuous data sets are also shown in Table 3. Our proposed method showed better performance than PSOPRS, PSOPRSN ( α=0.9 and α=0.5, where α is a parameter crossroads to the degree of dependency) PSOPRSE in every data set. Although the other methods, PSOFRFSA, PSOFRFSAN, and PSOFRFSANA, make reasonable improvements over our suggested approach in some data sets, it is not always the case. For instance, our method indicated better accuracy in the Musk-Version1 data set. Bold font indicates the best result for each data set based on the error rate.

Discussion

Feature selection has become an essential aspect of matured machine learning methods. Feature selection is also known as variable selection, feature reduction, attribute selection, or variable subset selection (Liu & Yu, 2005). This process is essential in practice for many reasons, especially if we have to collect data from costly sources such as sensors, patients, blood samples, etc. In such situations, we have to limit the number of features to a reasonable value; identifying the most important feature subset is crucial. Not only that but having fewer features also increases the computational efficiency and the prediction performances of the model. As a solution, we have proposed a new approach for the existing wrapper methods to select a minimal number of important features with similar performance. Hence, this is an important contribution as it reduces costs, especially in data collection.

Most of the wrapper feature selection methods compare scores of several feature subsets and select the one that gives the maximum score. There are other selections of fewer features with lower-score, yet with little difference in score. This article proposes and applies an extended version of selecting a minimal number of features subset instead of having the subset with the maximum score. PCLFS uses the sum of absolute values of principal component loadings to rank features and capture the most informative feature subset. It obtains the best feature subset by comparing the scores, where the feature subset that gives the best score is identified as the optimal feature subset. Still, some other feature subsets practically reduce the number of features with minimal score loss.

Our proposed method assesses the number of features below the maximum and receives the most beneficial smallest number of features and the feature subset with a tolerable score deduction. For the extended version, we only consider the feature subsets smaller than the previous subset selection; therefore, having a minimal feature set is guaranteed by the proposed approach under the threshold. The threshold plays a vital role in the introduced algorithm as the numerator, the maximum tolerable F1-score, is decided by the user using their domain knowledge and desire. The selection of the threshold is sensitive to the imbalance rate of the data. We can use a relatively larger threshold for highly imbalanced data to achieve a similar result. Although we have considered only five classification models in examples, like in PCLFS, the proposed method can also be fitted on any classification model as feature ordering is entirely independent of the classification method (Matharaarachchi, Domaratzki & Muthukumarana, 2021).

Although the underlying truth of the real-world data is hidden, we compare the result of the proposed method with existing PCLFS, RFE, and some other PSO-based feature selection methods on the same real data sets to compare the accuracy of each method.

Conclusion

This study introduces a novel gradient-based algorithm to further reduce the number of features with a similar performance to existing greedy feature selection approaches. The extended version of the existing PCLFS method was implemented to identify the most informative features first. First, we compare the proposed approach to PCLFS and RFE results on simulated data sets. and real-world data sets. Simulation results clearly shown that the proposed method makes a reasonable improvement over existing results, especially when we have a balanced data set and large sample size. For this purpose, we can re-balance the data set using existing methods such as SMOTE (Chawla et al., 2002). Then the results were compared with the most commonly used RFE method and some other PSO-based feature selection techniques for different continuous data sets. The results show that the proposed method allows us to select a subset that is often significantly smaller than that chosen by the original PCLFS method. A smaller informative feature set enables faster processing of data with higher accuracy, especially as more computationally expensive classification methods are used.

A simulation results for different classification models

Referring to the Simulation Results Section, Figs. A1–A4 present the results of the comparison of RFE, PCLFS, and PCLFS-ext methods for other classification models such as LGBM_C, Decision Tree, RFC and SVM_Linear with highly imbalanced data with 90%:10% rate and a sample size of 1,000. As discussed in “Simulation Results”, it is observed that, other than having higher model F1-scores and feature selection correct percentages, PCLFS-ext method also selects a lower number of features for many choices of informative features than the RFE method.

A.1 Light gradient boosting (LGBM_C)

Figure A1 Simulation results for LGBM_C-1,000 sample size.

Rows represent final F1-scores, Feature selection correct percentages, and the number of informative selected features, whereas the left-hand side column with original data and right is with SMOTE data for the Lgbm_C with a threshold of 0.0017.

A.2 Decision trees

Figure A2 Simulation results for Decision Trees-1,000 sample size.

Rows represent final F1-scores, Feature selection correct percentages, and the number of informative selected features, whereas the left-hand side column with original data and right is with SMOTE data for the Decision tree classifier with a threshold of 0.0017.

A.3 Random forest classifier (RFC)

Figure A3 Simulation results for RFC-1,000 sample size.

Rows represent final F1-scores, Feature selection correct percentages, and the number of informative selected features, whereas the left-hand side column with original data and right is with SMOTE data for the RFC with a threshold of 0.0017.

A.4 SVM linear

Figure A4 Simulation results for SVM_Linear-1,000 sample size.

Rows represent final F1-scores, Feature selection correct percentages, and the number of informative selected features, whereas the left-hand side column with original data and right is with SMOTE data for the SVM-linear classifier with a threshold of 0.0017.

Supplemental Information

Supplemental Information 1 Finding the minimal number of features with similar performance by introducing an extended version of the PCLFS method.

main.py runs the simulation study and application data where Grid_Score_Calc.py contains the introduced algorithm for the extended version of the PCLFS method. The SPECT.csv file is a publicly available Single-photon emission computed tomography (SPECT) heart data set, which describes diagnosing cardiac abnormalities using SPECT. Response of the data set consists of two categories: normal (0) and abnormal (1), by considering the diagnosis of images. This data consists of binary class imbalanced data with a higher number of numerical features and a lower number of instances.

Click here for additional data file.

Supplemental Information 2 Simulation results for different classification models.

Click here for additional data file.

Additional Information and Declarations

Competing Interests

Author Contributions

Data Availability

The authors declare that they have no competing interests.

Surani Matharaarachchi conceived and designed the experiments, performed the experiments, analyzed the data, performed the computation work, prepared figures and/or tables, authored or reviewed drafts of the article, and approved the final draft.

Mike Domaratzki conceived and designed the experiments, authored or reviewed drafts of the article, and approved the final draft.

Saman Muthukumarana conceived and designed the experiments, authored or reviewed drafts of the article, and approved the final draft.

The following information was supplied regarding data availability:

The code files are available at GitHub: https://github.com/SuraniM/PCLFS_Extended.

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
