# Peer review of "Minimizing features while maintaining performance in data classification problems"

_PeerJ Computer Science, doi:10.7717/peerj-cs.1081_

## Round 0.1 · original submission · Major Revisions

The reviewers have raised some concerns on the paper. A major revision is needed before further processing. Please provide a detailed response letter. Thanks.

·

Basic reporting

In this paper, authors proposed an extended version of their feature selection method (PCLFS) proposed in 2021.
This paper is well organized and well written and authors tried to improve the previous FS method PCLFS.
BUT authors are advised to carefully go through the manuscript and revised some points:

• Authors should give a title to Figure 1 then add the explanation of the blue point and the red point… Also change the label of x axis (Number of selected features). Same thing for Fig9, 10, 11, 12.
• Authors are advised to add a Conclusion Section in their paper; the Discussion Section here can be divided and of course authors should give more details and analysis of their results in Discussion section and add perspectives in Conclusion Section.
• Table 2 is not well done!!

Experimental design

no comment

Validity of the findings

• Authors have to compare their work with some new state of the art methods (2021, 2022)
• Authors are advised to add a Conclusion Section in their paper; the Discussion Section here can be divided and of course authors should give more details and analysis of their results in Discussion section and add perspectives in Conclusion Section.
• Authors have to add a link to download data (SPECTF heart data) used in this paper.
• To be more convincing, authors are invited to test their method on much more high dimensional data especially on real datasets. Authors can access to kaggle or UCI Machine Learning Repository to download real datasets (two classes).
• Authors are invited to compare also their work with other interesting and new FS methods for eg. the FS method using PSO and fuzzy rough set by Huda and Banka published in Soft Computing in 2022. Authors can discover the work of A Hashemi et al. published in the International Journal of Machine Learning and Cybernetics published in 2022.

Additional comments

• Line 77: authors can add ‘the remainder of this paper ..’
• Try to put in bold the best results in the table 1.

·

Basic reporting

1. The introduction section needs some more detail. For example, it misses a survey for recent feature selection methods that scores groups of features to reduce the number of selected features such as SVM-RCE (See ref 1 and 2).

ref1) Yousef, M., Jung, S., Showe, L.C. et al. Recursive Cluster Elimination (RCE) for classification and feature selection from gene expression data. BMC Bioinformatics 8, 144 (2007). https://doi.org/10.1186/1471-2105-8-144
ref2) Yousef M, Bakir-Gungor B, Jabeer A, Goy G, Qureshi R, C Showe L. Recursive Cluster Elimination based Rank Function (SVM-RCE-R) implemented in KNIME. F1000Res. 2020;9:1255. Published 2020 Oct 19. doi:10.12688/f1000research.26880.2

2. I would suggest making the code publicly available to allow others to validate the results.

Experimental design

The authors propose PCLFS-ext, which is an extension of an existing feature selection approach, named PCLFS. PCLFS-ext selects a smaller feature subset with similar performance to the original subset under a pre-defined threshold. I have the following concerns:

1. The merit of the suggested approach is not clearly explained. Why this approach should work well?

2. The suggested method requires being explained in more detail. I would like to see a specific example of how this approach is working.

3. The implementation details are missing for the suggested method and for the other algorithms. The authors need to provide the specific parameters used in each used algorithm.

Validity of the findings

Based on the presented results, it seems that PCLFS-ext is reaching its aims. However, I have the following concerns:

1. I am not sure why the authors decided to use imbalanced data and produce many results using SMOTE. I would like to see more results with different datasets such as gene expression and other big data in terms of features.

2. It seems to me that Table2 is not an informative table. Why do the authors need to present all those features' names? The authors might present just statistics about it in order to support their claim.

Additional comments

NA

---

## Round 0.2 · Minor Revisions

The reviewer still has some lingering comments. Please address them in the revised version. Thanks.

·

Basic reporting

A considerable work done in this revised paper. Minor modifications can improve this paper.

1. When we add a figure in a paper, we should add in the legend “Figure” its title then a description if there is ... For example:
Figure1. Ideal feature selection using PCLFS method. Dotted line indicates the actual ….
-->So please correct all the figures in this paper.

2. Authors are advised to go through their paper and choose to cite the proposed extended method as “PCLFS-extended” or “PCLFS-ext”; means please put just one annotation in all the paper (in tables, figures, and text..)

3. Merge Table 3 and Table 4 in the same table. This can be more visible to compare the proposed method with PSO_based methods and please try always to highlight the best results.

Experimental design

--

Validity of the findings

--

Additional comments

--

---

## Round 0.3 · accepted · Accept

The paper can be accepted. Congratulations.

·

Basic reporting

_

Experimental design

__

Validity of the findings

__

Additional comments

__